# Label Efficient Learning of Transferable Representations across Domains and Tasks

**Zelun Luo**
Stanford University
zelunluo@stanford.edu

**Yuliang Zou**
Virginia Tech
ylzou@vt.edu

**Judy Hoffman**
University of California, Berkeley
jhoffman@eecs.berkeley.edu

**Li Fei-Fei**
Stanford University
feifeili@cs.stanford.edu

## Abstract

We propose a framework that learns a representation transferable across different domains and tasks in a label efficient manner. Our approach battles domain shift with a domain adversarial loss, and generalizes the embedding to novel task using a metric learning-based approach. Our model is simultaneously optimized on labeled source data and unlabeled or sparsely labeled data in the target domain. Our method shows compelling results on novel classes within a new domain even when only a few labeled examples per class are available, outperforming the prevalent fine-tuning approach. In addition, we demonstrate the effectiveness of our framework on the transfer learning task from image object recognition to video action recognition.

## 1 Introduction

Humans are exceptional visual learners capable of generalizing their learned knowledge to novel domains and concepts and capable of learning from few examples. In recent years, computational models based on end-to-end learnable convolutional networks have made significant improvements for visual recognition [18, 28, 54] and have been shown to demonstrate some cross-task generalizations [8, 48] while enabling faster learning of subsequent tasks as most frequently evidenced through fine-tuning [14, 36, 50].

However, most efforts focus on the supervised learning scenario where a closed world assumption is made at training time about both the domain of interest and the tasks to be learned. Thus, any generalization ability of these models is only an observed byproduct. There has been a large push in the research community to address generalizing and adapting deep models across different domains [64, 13, 58, 38], to learn tasks in a data efficient way through few shot learning [27, 70, 47, 11], and to generically transfer information across tasks [1, 14, 50, 35].

While most approaches consider each scenarios in isolation we aim to directly tackle the joint problem of adapting to a novel domain which has new tasks and few annotations. Given a large labeled source dataset with annotations for a task set, A, we seek to transfer knowledge to a sparsely labeled target domain with a possibly wholly new task set, B. This setting is in line with our intuition that we should be able to learn reusable and general purpose representations which enable faster learning of future tasks requiring less human intervention. In addition, this setting matches closely to the most common practical approach for training deep models which is to use a large labeled source dataset (often ImageNet [6, 52]) to train an initial representation and then to continue supervised learning with a new set of data and often with new concepts.

In our approach, we jointly adapt a source representation for use in a distinct target domain using a new multilayer unsupervised domain adversarial formulation while introducing a novel cross-domain and within domain class similarity objective. This new objective can be applied even when the target domain has non-overlapping classes to the source domain.

We evaluate our approach in the challenging setting of joint transfer across domains and tasks and demonstrate our ability to successfully transfer, reducing the need for annotated data for the target domain and tasks. We present results transferring from a subset of Google Street View House Numbers (SVHN) [41] containing only digits 0-4 to a subset of MNIST [29] containing only digits 5-9. Secondly, we present results on the challenging setting of adapting from ImageNet [6] object-centric images to UCF-101 [57] videos for action recognition.

## 2 Related work

**Domain adaptation.** Domain adaptation seeks to learn from related source domains a well performing model on target data distribution [4]. Existing work often assumes that both domains are defined on the same task and labeled data in target domain is sparse or non-existent [64]. Several methods have tackled the problem with the Maximum Mean Discrepancy (MMD) loss [17, 36, 37, 38, 73] between the source and target domain. Weight sharing of CNN parameters [58, 22, 21, 3] and minimizing the distribution discrepancy of network activations [51, 65, 30] have also shown convincing results. Adversarial generative models [33, 32, 2, 59] aim at generating source-like data with target data by training a generator and a discriminator simultaneously, while adversarial discriminative models [62, 64, 13, 12, 23] focus on aligning embedding feature representations of target domain to source domain. Inspired by adversarial discriminative models, we propose a method that aligns domain features with multi-layer information.

**Transfer learning.** Transfer learning aims to transfer knowledge by leveraging the existing labeled data of some related task or domain [45, 71]. In computer vision, examples of transfer learning include [1, 31, 61] which try to overcome the deficit of training samples for some categories by adapting classifiers trained for other categories [43]. With the power of deep supervised learning and the ImageNet dataset [6, 52], learned knowledge can even transfer to a totally different task (i.e. image classification $\rightarrow$ object detection [50, 49, 34]; image classification $\rightarrow$ semantic segmentation [35]) and then achieve state-of-the-art performance. In this paper, we focus on the setting where source and target domains have differing label spaces but the label spaces share the same structure. Namely adapting between classifying different category sets but not transferring from classification to a localization plus classification task.

**Few-shot learning.** Few-shot learning seeks to learn new concepts with only a few annotated examples. Deep siamese networks [27] are trained to rank similarity between examples. Matching networks [70] learns a network that maps a small labeled support set and an unlabeled example to its label. Aside from these metric learning-based methods, meta-learning has also served as a essential part. Ravi et al. [47] propose to learn a LSTM meta-learner to learn the update rule of a learner. Finn et al. [11] tries to find a good initialization point that can be easily fine-tune with new examples from new tasks. When there exists a domain shift, the results of prior few-shot learning methods are often degraded.

**Unsupervised learning.** Many unsupervised learning algorithms have focused on modeling raw data using reconstruction objectives [19, 69, 26]. Other probabilistic models include restricted Boltzmann machines [20], deep Boltzmann machines [53], GANs [15, 10, 9], and autoregressive models [42, 66] are also popular. An alternative approach, often terms "self-supervised learning" [5], defines a pretext task such as predicting patch ordering [7], frame ordering [40], motion dynamics [39], or colorization [72], as a form of indirect supervision. Compared to these approaches, our unsupervised learning method does not rely on exploiting the spatial or temporal structure of the data, and is therefore more generic.

## 3 Method

We introduce a semi-supervised learning algorithm which transfers information from a large labeled source domain, $\mathcal{S}$, to a sparsely labeled target domain, $\mathcal{T}$. The goal being to learn a strong target

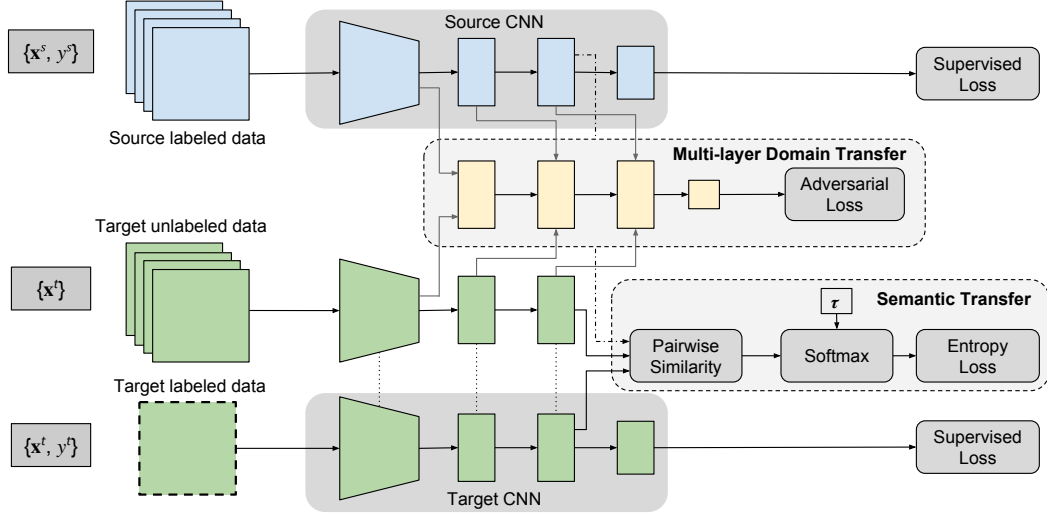

Figure 1: Our proposed learning framework for joint transfer across domains and semantic transfer across source and target and across target labeled to unlabeled data. We introduce a domain discriminator which aligns source and target representations across multiple layers of the network through domain adversarial learning. We enable semantic transfer through minimizing the entropy of the pairwise similarity between unlabeled and labeled target images and use the temperature of the softmax over the similarity vector to allow for non-overlapping label spaces.

classifier without requiring the large annotation overhead required for standard supervised learning approaches.

In fact, this setting is very commonly explored for convolutional network (convnet) based recognition methods. When learning with convnets the usual learning procedure is to use a very large labeled dataset (e.g. ImageNet [6, 52]) for initial training of the network parameters (termed pre-training). The learned weights are then used as initialization for continued learning on new data and for new tasks, called fine-tuning. Fine-tuning has been broadly applied to reduce the number of labeled examples needed for learning new tasks, such as recognizing new object categories after ImageNet pre-training [54, 18], or learning new label structures such as detection after classficiation pre-training [14, 50]. Here we focus on transfer in the case of a shared label structure (e.g. classification of different category sets).

We assume the source domain contains $n^s$ images, $\mathbf{x}^s \in \mathcal{X}^\mathcal{S}$, with associated labels, $\mathbf{y}^s \in \mathcal{Y}^\mathcal{S}$. Similarly, the target domain consists of $n^t$ unlabeled images, $\tilde{\mathbf{x}}^t \in \tilde{\mathcal{X}}^\mathcal{T}$, as well as $m^t$ images, $\mathbf{x}^t \in \mathcal{X}^\mathcal{T}$, with associated labels, $\mathbf{y}^t \in \mathcal{Y}^\mathcal{T}$. We assume that the target domain is only sparsely labeled so that the number of image-label pairs is much smaller than the number of unlabeled images, $m^t \ll n^t$. Additionally, the number of source labeled images is assumed to be much larger than the number of target labeled images, $m^t \ll n^s$.

Unlike standard domain adaptation approaches which transfer knowledge from source to target domains assuming a marginal or conditional distribution shift under a shared label space ($\mathcal{Y}^\mathcal{S} = \mathcal{Y}^\mathcal{T}$), we tackle joint image or feature space adaptation as well as transfer across semantic spaces. Namely, we consider the case where the source and target label spaces are not equal, $\mathcal{Y}^\mathcal{S} \neq \mathcal{Y}^\mathcal{T}$, and even the most challenging case where the sets are non-overlapping, $\mathcal{Y}^\mathcal{S} \cap \mathcal{Y}^\mathcal{T} = \emptyset$.

## 3.1 Joint domain and semantic transfer

Our approach consists of unsupervised feature alignment between source and target as well as semantic transfer to the unlabeled target data from either the labeled target or the labeled source data. We introduce a new multi-layer domain discriminator which can be used for domain alignment following the recent domain adversarial learning approaches [13, 64]. We next introduce a new semantic transfer learning objective which uses cross category similarity and can be tuned to account for varying size of label set overlap.

We depict our overall model in Figure 1. We take the $n^s$ source labeled examples, $\{\mathbf{x}^s, \mathbf{y}^s\}$, the $m^t$ target labeled examples, $\{\mathbf{x}^t, \mathbf{y}^t\}$, and the $n^t$ unlabeled target images, $\{\tilde{\mathbf{x}}^t\}$ as input. We learn an initial layered source representation and classification network (depicted in blue in Figure 1) using standard supervised techniques. We then initialize the target model (depicted in green in Figure 1) with the source parameters and begin our adaptive transfer learning.

Our model jointly optimizes over a target supervised loss, $\mathcal{L}_{\text{sup}}$, a domain transfer objective, $\mathcal{L}_{DT}$, and finally a semantic transfer objective, $\mathcal{L}_{ST}$. Thus, our total objective can be written as follows:

$$\mathcal{L}(\mathcal{X}^{\mathcal{S}}, \mathcal{Y}^{\mathcal{S}}, \mathcal{X}^{\mathcal{T}}, \mathcal{Y}^{\mathcal{T}}, \tilde{\mathcal{X}}^{\mathcal{T}}) = \mathcal{L}_{\text{sup}}(\mathcal{X}^{\mathcal{T}}, \mathcal{Y}^{\mathcal{T}}) + \alpha \mathcal{L}_{DT}(\mathcal{X}^{\mathcal{S}}, \tilde{\mathcal{X}}^{\mathcal{T}}) + \beta \mathcal{L}_{ST}(\mathcal{X}^{\mathcal{S}}, \mathcal{X}^{\mathcal{T}}, \tilde{\mathcal{X}}^{\mathcal{T}}) \quad (1)$$

where the hyperparameters $\alpha$ and $\beta$ determine the influence of the domain transfer loss and the semantic transfer loss, respectively. In the following sections we elaborate on our domain and semantic transfer objectives.

### 3.2 Multi-layer domain adversarial loss

We define a novel domain alignment objective function called *multi-layer domain adversarial loss*. Recent efforts in deep domain adaptation have shown strong performance using feature space domain adversarial objectives [13, 64]. These methods learn a target representation such that the target distribution viewed under this model is aligned with the source distribution viewed under the source representation. This alignment is accomplished through an adversarial minimization across domain, analogous to the prevalent generative adversarial approaches [15]. In particular, a domain discriminator, $D(\cdot)$, is trained to classify whether a particular data point arises from the source or the target domain. Simultaneously, the target embedding function $E^t(\mathbf{x}^t)$ (defined as the application of layers of the network is trained to generate the target representation that cannot be distinguished from the source domain representation by the domain discriminator. Similar to [63, 64], we consider a representation to be domain invariant if the domain discriminator can not distinguish examples from the two domains.

Prior work considers alignment for a single layer of the embedding at a time and as such learns a domain discriminator which takes the output from the corresponding source and target layers as input. Separately, domain alignment methods which focus on first and second order statistics have shown improved performance through applying domain alignment independently at multiple layers of the network [36]. Rather than learning independent discriminators for each layer of the network we propose a simultaneous alignment of multiple layers through a multi-layer discriminator.

At each layer of our multi-layer domain discriminator, information is accumulated from both the output from the previous discriminator layer as well as the source and target activations from the corresponding layer in their respective embeddings. Thus, the output of each discriminator layer is defined as:

$$\mathbf{d}_l = D_l(\sigma(\gamma \mathbf{d}_{l-1} \oplus E_l(\mathbf{x}))) \quad (2)$$

where $l$ is the current layer, $\sigma(\cdot)$ is the activation function, $\gamma \leq 1$ is the decay factor, $\oplus$ represents concatenation or element-wise summation, and $\mathbf{x}$ is taken either from source data $\mathbf{x}^s \in \mathcal{X}^{\mathcal{S}}$, or target data $\tilde{\mathbf{x}}^t \in \tilde{\mathcal{X}}^{\mathcal{T}}$. Notice that the intermediate discriminator layers share the same structure with their corresponding encoding layers to match the dimensions.

Thus, the following loss functions are proposed to optimize the multi-layer domain discriminator and the embeddings, respectively, according to our domain transfer objective:

$$\mathcal{L}_{DT}^{D} = -\mathbb{E}_{\mathbf{x}^s \sim \mathcal{X}^{\mathcal{S}}} \left[\log \mathbf{d}_l^s\right] - \mathbb{E}_{\mathbf{x}^t \sim \mathcal{X}^{\mathcal{T}}} \left[\log(1 - \mathbf{d}_l^t)\right] \quad (3)$$

$$\mathcal{L}_{DT}^{E^t} = -\mathbb{E}_{\mathbf{x}^s \sim \mathcal{X}^{\mathcal{S}}} \left[\log(1 - \mathbf{d}_l^s)\right] - \mathbb{E}_{\mathbf{x}^t \sim \mathcal{X}^{\mathcal{T}}} \left[\log \mathbf{d}_l^t\right] \quad (4)$$

where $\mathbf{d}_l^s, \mathbf{d}_l^t$ are the outputs of the last layer of the source and target multi-layer domain discriminator. Note that these losses are placed after the final domain discriminator layer and the last embedding layer but then produce gradients which back-propagate throughout all relevant lower layer parameters. These two losses together comprise $L_{DT}$, and there is no iterative optimization procedure involved.

This multi-layer discriminator (shown in Figure 1 - yellow) allows for deeper alignment of the source and target representations which we find empirically results in improved target classification performance as well as more stable adversarial learning.

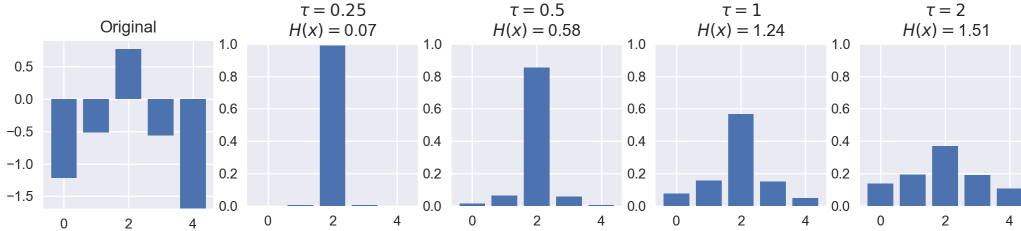

Figure 2: We illustrate the purpose of temperature ($\tau$) for our pairwise similarity vector. Consider an example target unlabeled point and its similarity to four labeled source points (*x-axis*). We show here, original unnormalized scores (*leftmost*) as well as the same similarity scores after applying softmax with different temperatures, $\tau$. Notice that entropy values, $H(x)$, have higher variance for scores normalized with a small temperature softmax.

### 3.3 Cross category similarity for semantic transfer

In the previous section, we introduced a method for transferring an embedding from the source to the target domain. However, this only enforces alignment of the global domain statistics with no class specific transfer. Here, we define a new semantic transfer objective, $\mathcal{L}_{ST}$, which transfers information from a labeled set of data to an unlabeled set of data by minimizing the entropy of the softmax with temperature of the similarity vector between an unlabeled point and all labeled points. Thus, this loss may be applied either between the source and unlabeled target data or between the labeled and unlabeled target data.

For each unlabeled target image, $\tilde{\mathbf{x}}^t$, we compute the similarity, $\psi(\cdot)$, to each labeled example or to each prototypical example [56] per class in the labeled set. For simplicity of presentation let us consider semantic transfer from the source to the target domain first. For each target unlabeled image we compute a similarity vector where the $i^{th}$ element is the similarity between this target image and the $i^{th}$ labeled source image: $[v_s(\tilde{\mathbf{x}}^t)]_i = \psi(\tilde{\mathbf{x}}^t, \mathbf{x}_i^s)$. Our semantic transfer loss can be defined as follows:

$$\mathcal{L}_{ST}(\tilde{\mathcal{X}^{\mathcal{T}}}, \mathcal{X}^{\mathcal{S}}) \quad = \quad \sum_{\tilde{\mathbf{x}}^t \in \tilde{\mathcal{X}^{\mathcal{T}}}} H(\sigma(v_s(\tilde{\mathbf{x}}^t)/\tau)) \tag{5}$$

where, $H(\cdot)$ is the information entropy function, $\sigma(\cdot)$ is the softmax function and $\tau$ is the temperature of the softmax. Note that the temperature can be used to directly control the percentage of source examples we expect the target example to be similar to (see Figure 2).

Entropy minimization has been widely used for unsupervised [44] and semi-supervised [16] learning by encouraging low density separation between clusters or classes. Recently this principle of entropy minimization has be applied for unsupervised adaptation [38]. Here, the source and target domains are assumed to share a label space and each unlabeled target example is passed through the initial source classifier and the entropy of the softmax output scores is minimized.

In contrast, we do not assume a shared label space between the source and target domains and as such can not assume that each target image maps to a single source label. Instead, we compute pairwise similarities between target points and the source points (or per class averages of source points [56]) across the features spaces aligned by our multi-layer domain adversarial transfer. We then tune the softmax temperature based on the expected similarity between the source and target labeled set. For example, if the source and target label set overlap, then a small temperature will encourage each target point to be very similar to one source class, whereas a larger temperature will allow for target points to be similar to multiple source classes.

For semantic transfer within the target domain, we utilize the metric-based cross entropy loss between labeled target examples to stabilize and improve the learning. For a labeled target example, in addition to the traditional cross entropy loss, we also calculate a metric-based cross entropy loss [1]. Assume we have $k$ labeled examples from each class in the target domain. We compute the embedding for

each example and then the centroid $c_i^{\mathcal{T}}$ of each class in the embedding space. Thus, we can compute the similarity vector for each labeled example, where the $i^{th}$ element is the similarity between this labeled example and the centroid of each class: $[v_t(\mathbf{x}^t)]_i = \psi(\mathbf{x}^t, c_i^{\mathcal{T}})$. We can then calculate the metric based cross entropy loss:

$$\mathcal{L}_{ST,\text{sup}}(\mathcal{X}^{\mathcal{T}}) = - \sum_{\{\mathbf{x}^t, \mathbf{y}^t\} \in \mathcal{X}^{\mathcal{T}}} \log \frac{\exp\left([v_t(\mathbf{x}^t)]_{\mathbf{y}^t}\right)}{\sum_{i=1}^{n} \exp\left([v_t(\mathbf{x}^t)]_i\right)} \qquad (6)$$

Similar to the source-to-target scenario, for target-to-target we also have the unsupervised part,

$$\mathcal{L}_{ST,\text{unsup}}(\tilde{\mathcal{X}}^{\mathcal{T}}, \mathcal{X}^{\mathcal{T}}) \;\; = \;\; \sum_{\tilde{\mathbf{x}}^t \in \tilde{\mathcal{X}}^{\mathcal{T}}} H(\sigma(v_t(\tilde{\mathbf{x}}^t)/\tau)) \qquad (7)$$

With the metric-based cross entropy loss, we introduce the constraint that the target domain data should be similar in the embedding space. Also, we find that this loss can provide a guidance for the unsupervised semantic transfer to learn in a more stable way. $\mathcal{L}_{ST}$ is the combination of $\mathcal{L}_{ST,\text{unsupervised}}$ from source-target (Equation 5), $\mathcal{L}_{ST,\text{supervised}}$ from source-target (Equation 6), and $\mathcal{L}_{ST,\text{unsupervised}}$ from target-target (Equation 7), i.e.,

$$\mathcal{L}_{ST}(\mathcal{X}^{\mathcal{S}}, \mathcal{X}^{\mathcal{T}}, \tilde{\mathcal{X}}^{\mathcal{T}}) = \mathcal{L}_{ST}(\tilde{\mathcal{X}}^{\mathcal{T}}, \mathcal{X}^{\mathcal{S}}) + \mathcal{L}_{ST,\text{sup}}(\mathcal{X}^{\mathcal{T}}) + \mathcal{L}_{ST,\text{unsup}}(\tilde{\mathcal{X}}^{\mathcal{T}}, \mathcal{X}^{\mathcal{T}}) \qquad (8)$$

## 4  Experiment

This section is structured as follows. In section 4.1, we show that our method outperform fine-tuning approach by a large margin, and all parts of our method are necessary. In section 4.2, we show that our method can be generalized to bigger datasets. In section 4.3, we show that our multi-layer domain adversarial method outperforms state-of-the-art domain adversarial approaches.

**Datasets** We perform adaptation experiments across two different paired data settings. First for adaptation across different digit domains we use MNIST [29] and Google Street View House Numbers (SVHN) [41]. The MNIST handwritten digits database has a training set of 60,000 examples, and a test set of 10,000 examples. The digits have been size-normalized and centered in fixed-size images. SVHN is a real-world image dataset for machine learning and object recognition algorithms with minimal requirement on data preprocessing and formatting. It has 73257 digits for training, 26032 digits for testing. As our second experimental setup, we consider adaptation from object centric images in ImageNet [52] to action recognition in video using the UCF-101 [57] dataset. ImageNet is a large benchmark for the object classification task. We use the task 1 split from ILSVRC2012. UCF-101 is an action recognition dataset collected on YouTube. With 13,320 videos from 101 action categories, UCF-101 provides a large diversity in terms of actions and with the presence of large variations in camera motion, object appearance and pose, object scale, viewpoint, cluttered background, illumination conditions, etc.

**Implementation details** We pre-train the source domain embedding function with cross-entropy loss. For domain adversarial loss, the discriminator takes the last three layer activations as input when the number of output classes are the same for source and target tasks, and takes the second last and third last layer activations when they are different. The similarity score is chosen as the dot product of the normalized support features and the unnormalized target feature. We use the temperature $\tau = 2$ for source-target semantic transfer and $\tau = 1$ for within target transfer as the label space is shared. We use $\alpha = 0.1$ and $\beta = 0.1$ in our objective function. The network is trained with Adam optimizer [25] and with learning rate $10^{-3}$. We conduct all the experiments with the PyTorch framework.

### 4.1  SVHN 0-4 $\rightarrow$ MNIST 5-9

**Experimental setting.** In this experiment, we define three datasets: (i) labeled data in source domain $\mathcal{D}_1$; (ii) few labeled data in target domain $\mathcal{D}_2$; (iii) unlabeled data in target domain $\mathcal{D}_3$. We take the training split of SVHN dataset as dataset $\mathcal{D}_1$. To fairly compare with traditional learning paradigm and episodic training, we subsample $k$ examples from each class to construct dataset $\mathcal{D}_2$ so that we can perform traditional training or episodic $(k-1)$-shot learning. We experiment with $k = 2, 3, 4, 5$, which corresponds to $10, 15, 20, 25$ labeled examples, or $0.017\%, 0.025\%, 0.333\%, 0.043\%$ of the

total training data respectively. Since our approach involves using annotations from a small subset of the data, we randomly subsample 10 different subsets $\{\mathcal{D}_2^i\}_{i=1}^{10}$ from the training split of MNIST dataset, and use the remaining data as $\{\mathcal{D}_3^i\}_{i=1}^{10}$ for each $k$. Note that source domain and target domain have non-overlapping classes: we only utilize digits 0-4 in SVHN, and digits 5-9 in MNIST.

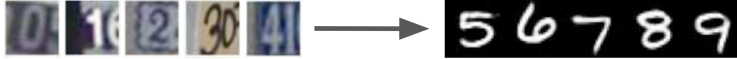

Figure 3: An illustration of our task. Our model effectively transfer the learned representation on SVHN digits 0-4 (left) to MNIST digits 5-9 (right).

**Baselines and prior work.** We compare against six different methods: (i) *Target only*: the model is trained on $\mathcal{D}_2$ from scratch; (ii) *Fine-tune*: the model is pretrained on $\mathcal{D}_1$ and fine-tuned on $\mathcal{D}_2$; (iii) *Matching networks* [70]: we first pretrain the model on $\mathcal{D}_3$, then use $\mathcal{D}_2$ as the support set in the matching networks; (iv) *Fine-tuned matching networks*: same as baseline iii, except that for each $k$ the model is fine-tuned on $\mathcal{D}_2$ with 5-way $(k-1)$-shot learning: $k-1$ examples in each class are randomly selected as the support set, and the last example in each class is used as the query set; (v) *Fine-tune + adversarial*: in addition to baseline ii, the model is also trained on $\mathcal{D}_1$ and $\mathcal{D}_3$ with a domain adversarial loss; (vi.) *Full model*: fine-tune the model with the proposed multi-layer domain adversarial loss.

**Results and analysis.** We calculate the mean and standard error of the accuracies across 10 sets of data, which is shown in Table 1. Due to domain shift, matching networks perform poorly without fine-tuning, and fine-tuning is only marginally better than training from scratch. Our method with multi-layer adversarial only improves the overall performance, but is more sensitive to the subsampled data. Our method achieves significant performance gain, especially when the number of labeled examples is small ($k = 2$). For reference, fine-tuning on full target dataset gives an accuracy of 99.65%.

Table 1: The test accuracies of the baseline models and our method. Row 1 to row 6 correspond (in the same order) to the six methods proposed in section 4.2. Note that the accuracies of two matching net methods are calculated based on nearest neighbors in the support set. We report the mean and the standard error of each method across 10 different subsampled data.

| Method | k=2 | k=3 | k=4 | k=5 |
|---|---|---|---|---|
| Target only | $0.642 \pm 0.026$ | $0.771 \pm 0.015$ | $0.801 \pm 0.010$ | $0.840 \pm 0.013$ |
| Fine-tune | $0.612 \pm 0.020$ | $0.779 \pm 0.018$ | $0.802 \pm 0.016$ | $0.830 \pm 0.011$ |
| Matching nets [70] | $0.469 \pm 0.019$ | $0.455 \pm 0.014$ | $0.566 \pm 0.013$ | $0.513 \pm 0.023$ |
| Fine-tuned matching nets | $0.645 \pm 0.019$ | $0.755 \pm 0.024$ | $0.793 \pm 0.013$ | $0.827 \pm 0.011$ |
| Ours: fine-tune + adv. | $0.702 \pm 0.020$ | $0.800 \pm 0.013$ | $0.804 \pm 0.014$ | $0.831 \pm 0.013$ |
| Ours: full model ($\gamma = 0.1$) | $\mathbf{0.917 \pm 0.007}$ | $\mathbf{0.936 \pm 0.006}$ | $\mathbf{0.942 \pm 0.006}$ | $\mathbf{0.950 \pm 0.004}$ |

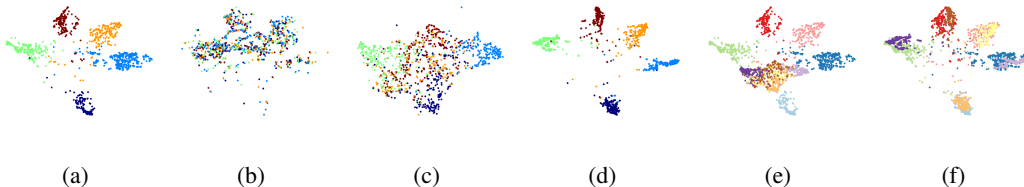

|   (a)   |   (b)   |   (c)   |   (d)   |   (e)   |   (f)   |

Figure 4: The t-SNE [68, 67] visualization of different feature embeddings. (a) Source domain embedding. (b) Target domain embedding using encoder trained with source domain domain. (c) Target domain embedding using encoder fine-tuned with target domain data. (d) Target domain embedding using encoder trained with our method. (e) An overlap of a and c. (f) An overlap of a and d. (best viewed in color and with zoom)

## 4.2 Image object recognition → video action recognition

**Problem analysis.** Many recent works [60, 24] study the domain shift between images and video in the object detection settings. Compared to still images, videos provide several advantages: (i) motion provides information for foreground vs background segmentation [46]; (ii) videos often show multiple views and thus provide 3D information. On the other hand, video frames usually suffer from: (i) motion blur; (ii) compression artifacts; (iii) objects out-of-focus or out-of-frame.

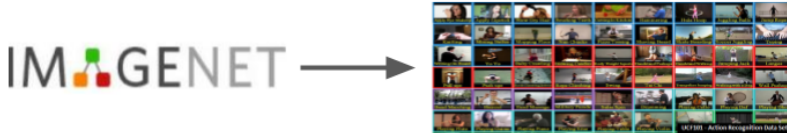

**Experimental setting.** In this experiment, we focus on three dataset splits: (i) ImageNet training set as the labeled data in source domain $\mathcal{D}_1$; (ii) $k$ video clips per class randomly sampled from UCF-101 training as the few labeled data in target domain set $\mathcal{D}_2$; (iii) the remaining videos in UCF-101 training set as the unlabeled data in target domain $\mathcal{D}_3$. We experiment with $k = 3, 5, 10$, which corresponds $303, 505, 1010$ video clips, or $2.27\%, 3.79\%, 7.59\%$ of the total training data respectively. Each experiment is run 3 times on $\mathcal{D}_1$, $\{\mathcal{D}_2^i\}_{i=1}^3$, and $\{\mathcal{D}_3^i\}_{i=1}^3$.

**Baselines and prior work.** We compare our method with two baseline methods: (i) *Target only*: the model is trained on $\mathcal{D}_2$ from scratch; (ii) *Fine-tune*: the model is first pre-trained on $\mathcal{D}_1$, then fine-tuned on $\mathcal{D}_2$. For reference, we report the performance of a fully supervised method [55].

**Results and analysis.** The accuracy of each model is shown in Table 2. We also fine-tune a model with all the labeled data for comparison. Per-frame performance (img) and average-across-frame performance (vid) are both reported. Note that we calculate the average-across-frame performance by averaging the *softmax* score of each frame in a video. Our method achieves significant improvement on average-across-frame performance over standard fine-tuning for each value of $k$. Note that compared to fine-tuning, our method has a bigger gap between per-frame and per-video accuracy. We believe that this is due to the semantic transfer: our entropy loss encourages a sharper softmax variance among per-frame softmax scores per video (if the variance is zero, then per-frame accuracy = per-video accuracy). By making more confident predictions among key frames, our method achieves a more significant gain with respective to per-video performance, even when there is little change in the per-frame prediction.

Table 2: Accuracy of UCF-101 action classification. The results of the two-stream spatial model are taken from [55] and vary depending on hyperparameters. We report the mean and the standard error of each method across 3 different subsampled data.

| Method | k=3 | k=5 | k=10 | All |
|---|---|---|---|---|
| Target only (img) | 0.098±0.003 | 0.126±0.022 | 0.100±0.035 | - |
| Target only (vid) | 0.105±0.003 | 0.133±0.024 | 0.106±0.038 | - |
| Fine-tune (img) | 0.380±0.013 | 0.486±0.012 | 0.529±0.039 | 0.672 |
| Fine-tune (vid) | 0.406±0.015 | 0.523±0.010 | 0.568±0.042 | 0.714 |
| Two-stream spatial [55] | - | - | - | 0.708 - 0.720 |
| Ours (img) | 0.393±0.006 | 0.459±0.013 | 0.523±0.002 | - |
| Ours (vid) | **0.467±0.007** | **0.545±0.014** | **0.620±0.005** | - |

## 4.3 Ablation: unsupervised domain adaptation

To validate our multi-layer domain adversarial loss objective, we conduct an ablation experiment for unsupervised domain adaptation. We compare against multiple recent domain adversarial unsupervised adaptation methods. In this experiment, we first pretrain a source embedding CNN on the training split SVHN [41] and then adapt the target embedding for MNIST by performing adversarial domain adaptation. We evaluate the classification performance on the test split of MNIST [29]. We follow the same training strategy and model architecture for the embedding network as [64].

All the models here have a two-step training strategy and share the first stage. ADDA [64] optimizes encoder and classifier simultaneously. We also propose a similar method, but optimize encoder only. Only we try a model with no classifier in the last layer (i.e. perform domain adversarial training in feature space). We choose $\gamma = 0.1$ as the decay factor for this model.

The accuracy of each model is shown in Table 3. We find that our method achieve $6.5\%$ performance gain over the best competing domain adversarial approach indicating that our multilayer objective indeed contributes to our overall performance. In addition, in our experiments, we found that the multilayer approach improved overall optimization stability, as evidenced in our small standard error.

Table 3: Experimental results on unsupervised domain adaptation from SVHN to MNIST. Results of Gradient reversal, Domain confusion, and ADDA are from [64], and the results of other methods are from experiments across 5 different subsampled data.

| Method | Accuracy |
|---|---|
| Source only | $0.601 \pm 0.011$ |
| Gradient reversal [13] | $0.739$ |
| Domain confusion [62] | $0.681 \pm 0.003$ |
| ADDA [64] | $0.760 \pm 0.018$ |
| Ours | $\mathbf{0.810 \pm 0.003}$ |

## 5 Conclusion

In this paper, we propose a method to learn a representation that is transferable across different domains and tasks in a data efficient manner. The framework is trained jointly to minimize the domain shift, to transfer knowledge to new task, and to learn from large amounts of unlabeled data. We show superior performance over the popular fine-tuning approach. We hope to keep improving the method in future work.

## Acknowledgement

We would like to start by thanking our sponsors: Stanford Computer Science Department and Stanford Program in AI-assisted Care (PAC). Next, we specially thank De-An Huang, Kenji Hata, Serena Yeung, Ozan Sener and all the members of Stanford Vision and Learning Lab for their insightful discussion and feedback. Lastly, we thank all the anonymous reviewers for their valuable comments.

## Footnotes

[1]We refer this as "metric-based" to cue the reader that this is not a cross entropy within the label space.

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
