[Reviews · NeurIPS 2017]

Reviewer 1



The works proposed a new framework to learn a representation that can transfer across domains as well as different tasks. There are two main novelties of this work: 1) a multi-layer domain adversarial loss vs existing work which learns a layer-by-layer embedding (generator) and discriminator; 2) a semantic transfer loss defined in equation (4) to address the cases when the source and target domain have different tasks (classes). Experimental results demonstrated that the multi-layer domain adversarial loss improves domain alignment as well as stability of adversarial training. Table 1 suggests that semantic transfer offers a significant boost to the performance of the proposed method. The paper is mostly well-written and easy to follow. The notations are not very consistent. Section 3.3 could be improved. The experimental results showed significant improvement over existing methods with the two modification introduced. There are three components introduced in the semantic transfer section. It is not clear from the experimental results that which one is contributing to the improvement mostly. It is not clear why the authors refer to the loss in equation (5) as a metric-based loss. In section 3.3, it seems like the similarity \phi(,) should be defined over the learned embedding instead of the original raw input? Please clarify the notation in section 3.3 to be consistent with section 3.2 if it is the case. In equation (4) and (6), do the authors use all the labeled examples or only the prototype examples to compute the cross-entropy? The former case seems prohibitively expensive, especially when the number of samples in the source domain is big. If it's the later case, how were the prototype examples chosen? Minor question: 1) what is the \gamma parameter shown in table 1 ?

Reviewer 2



The work presents a new method for knowledge transfer in neural network. In the setting studied here there is a source learning problem for which a large labeled training set exists however for the target task there exists only a small labeled training set and a larger unlabeled training set. Two main ideas are used to perform the knowledge transfer. First, a “translation” network is trained to convert the representation learned for the source task to the representation that is being learned for the target task using adversarial techniques. Second, entropy based loss is used to force the target representation to preserve class information. The method is tested on two tasks and shows very impressive gains. I find this paper well written and easy to follow. He authors present well the intuition behind their choices. The problem being studied is of great importance and the empirical work shows great improvements compared to the benchmark. Some minor comments: Line 23: “across across” --> “across” Line 69: "update rule of learner" --> "update rule of a learner" Paragraph beginning in line 185: would the use of the average example work when the classes are made of not connected components? For example, if you are trying to distinguish the even numbers from the odd numbers in the MNIST dataset. Paragraph beginning in line 214: how were these parameters selected? Did you perform any parameter sweep to tune them? Are they generic in the sense that you expect these values to work well for any task? Line 278: The sentence beginning with “Only we …” is unclear Line 279: the gain is 5 percentage points which is 6.5% in this case since 0.81/0.76 =~ 1.065

Reviewer 3



Nice paper on how we leverage representation learning to learn domain transfer and semantic transfer. I particularly like the way they separate the semantic layer for the target domain and the alignment of representations between the domains. One quick question is on why in the domain transfer layer we considered source labeled data and target unlabeled data. Target labeled data should also be leveraged to learn a better domain transfer. The experiments are convincing to me with good details and intuitive visualizations.

Reviewer 4



The authors of the manuscript propose a method for a challenging modification of the standard domain adaptation problem, where the label sets in the source and in the target domains do not have to be same (though, it is assumed that they are of the same kind, for example, both are classification problems). The first novelty of the proposed method is a different, multi-layer, domain adversarial loss that takes into account data representation at multiple layers of the network. I think description of this loss misses a few details: - (2) and (3) are not optimization problems and are formulated only for one layer - how are losses for multiple layers combined? - how do (2) and (3) contribute to L_{DT}? there is some iterative optimization procedure involved? The second novelty is a semantic transfer loss that facilitates information transfer between labeled and unlabelled data across domains. - why L_{ST,supervised} is not using the entropy (with small temperature)? - how are all three versions of L_{ST} combined together? - it is not quite clear to me which part of L_{ST} is resulting in improvements in the experiments, in particular, how important (and why) is source-to-target transfer, especially when the label sets are disjoint Additional comments: - do the authors have any intuition why the proposed method does not provide a (significant) improvement in the second experiments wth respect to per-frame performance? - matching between Table 1 and description of the baselines is unclear - I couldn’t find entropy minimisation in [37] (mentioned in line 174) Overall, I find the considered setting interesting and the experimental results promising, however, some clarification of the missing details of the method is needed for final evaluation